# Elaboration of Screening Scales for Mental Development Problems Detection in Russian Preschool Children: Psychometric Approach

**DOI:** 10.3390/diagnostics10090646

**Published:** 2020-08-28

**Authors:** Andrey Nasledov, Sergey Miroshnikov, Liubov Tkacheva, Vadim Goncharov

**Affiliations:** Department of Pedagogy and Pedagogical Psychology, Saint Petersburg State University, 7/7 Universitetskaya Emb., 199034 Saint Petersburg, Russia; andrey.nasledov@gmail.com (A.N.); sergeyamir@gmail.com (S.M.); wadimag@web.de (V.G.)

**Keywords:** screening, markers of mental development, ASD, developmental delay, construction of scales, 3-year-olds

## Abstract

Background: computer-based screenings are usually used for early detection of a child’s mental development problems. However, there are no such screenings in Russia yet. This study aimed to elaborate scales for rapid monitoring of mental development of 3-year-olds. Methods: 863 children took part in the study, among them 814 children of the group Norm, 49 children with developmental delay (DD), including 23 children with symptoms of autistic spectrum disorder (ASD). The multifactor study of mental development tool was used as a part of a software complex for longitudinal research for data collection. This study used a set of 233 tasks that were adequate for 3-year-olds. Exploratory and confirmatory factor analysis was used for the elaboration and factor validation of the scales. The structure of the relationship between scales and age was refined using structural equation modeling. Results: as a result of the research, screening scales were elaborated: “Logical reasoning”, “Motor skills”, “General awareness”, “Executive functions”. The factor validity and reliability of scales were proved. The high discriminability of the scales in distinguishing the “Norm” and “DD” samples was revealed. The developed test norms take into account the child’s age in days and allow identifying a “risk group” with an expected forecast accuracy of at least 90%. The obtained scales meet psychometric requirements for their application and allow creating an online screening system for wide application.

## 1. Introduction

To date, there are no doubts that the earlier a child’s problems in the cognitive and psychomotor development are detected, the higher the chances for timely intervention and changing the potential trajectory of his/her development. A temporary delay in the acquisition of cognitive, sensory and motor skills can be caused by various neurodevelopmental disorders [1], and quite often is burdened by autistic-like symptoms [2]. The problem of developmental delay (DD) at an early age needs precise consideration, since such children are at higher risk for mental disorders, compared with typically developing peers [3]. Also, behavioral disorders affect about 50% of children with DD or at risk for DD at the age of 2–3 years [4], and this tendency worsens over time if there is the absence of appropriate corrective interventions [5]. Burdening DD with autistic-like symptoms is a prognostic adverse circumstance. Symptoms of autistic spectrum disorder (ASD) are diverse and most often include various patterns of sensory disintegration, propensities to stereotyped movements and a limited behavioral repertoire. The task of diagnosing and quantifying children with ASD is difficult to implement primarily due to heterogeneity of symptoms, the absence of biological diagnostic markers and changes in diagnostic criteria [6]. A progress in early detection symptoms of ASD and following intervention can significantly improve the social adaptation of affected children. Therefore, more research has recently been devoted to the study of the predictors of ASD outcomes and the actual needs of adolescents and adults with autism [7,8].

Screening tools developed in the West for children with DD and ASD differ in their methodology. In the first case, such screenings of children’s mental development as Griffiths, Denver, Vineland are widely used [9,10]. To identify ASD, questionnaires and observation sheets for cognitive and adaptive behavior are mainly used, for example, ADOS (Autism Diagnostic Observation Schedule) [11] and ADI-R (Autism Diagnostic Interview-Revised) [12]. In Russia, however, there is no diagnostic screening developed for a sample of Russian children for early identification of problems or deviations in mental development. The complexity of translation and adaptation of Western methods is comparable to the elaboration of a new diagnostic tool, while the problem of taking into account the specifics of Russian children’s cognitive development is not solved, not to mention the cost of a license to use foreign screening. Therefore, we set out to create screening scales for early detection of possible problems in the cognitive and psychomotor development of children and differentiation of children into typically developing and ones who need the attention of specialists. Utilizing these early screening diagnostics, which take up to 20–30 min and can be carried out both at preschool educational institutions and at district clinics, we plan to optimize the process of early intervention and “unload” specialists, giving them the opportunity to work with children who really need special attention.

In one of the most well-known and widely used Western screenings, Denver [9], child development is evaluated by the following domains, implying that these areas are development vectors and related to the factor structure of intelligence: (1) large and fine motor skills; (2) speech development; and (3) communication and social adaptation. Similar scales were identified earlier in the elaboration of screening tools for early diagnosis of DD, and included 3 scales for 4–5-year-olds: “Logical reasoning”, “Motor skills” and “General awareness”. These scales coincided in content for children of these age groups, but differed in the task sets: for 4-year-olds - simpler, for 5-year-olds - more complex) [13].

In this study, we aimed to develop analogous scales for 3-year-olds. We paid attention to this age last because of the difficulties in data collection for samples of children with DD and ASD, since children in Russia only begin to attend preschool institutions at this age. Respectively, it is only from this age that children with developmental problems come to the attention of specialists and may be assessed as belonging to DD or ASD groups. In this regard, over the past 4 years of the study, the sample of 3-year-olds with established developmental problems consisted of only 26 children with DD and 23 children with ASD, which is almost 3 times less than the number of children in older groups. Due to the small sample sizes of children with DD and ASD in this study, in contrast to the previous ones, the development of scales was carried out on a sample of children without a diagnosis (belonging to Norm group), and data on children with DD and ASD was used to test the discriminative validity of the selected scales. However, we assumed that our research for 3-year-olds would reveal scales similar to those we obtained for older children, correlating with Denver screening, and including at least such factors as “General awareness”, “Motor skills”, and “Logical reasoning”.

## 2. Materials and Methods

In total, 840 3-year-olds were examined, evenly represented in the age range from 1095 to 1459 calendar days. Of these, the main sample consisted of 814 typically developing children (group “Norm”). A sample of 26 children with DD was used to test the discriminative validity of the developed scales. A sample of 23 children with ASD was used to analyze the applicability of the existing tasks bank (tasks, formalized observations) and determine the need to supplement to the tasks bank, as well as check the functionality of the software, taking into account the specialists’ feedbacks. Diagnosis of the children (including assignment to the Norm, DD or ASD groups) was carried out by experienced specialists from psychological counseling centers and pre-school institutions with the participation of neurologists, psychologists, psychiatrists before the survey. Data collection was carried out by tutors and psychologists engaged in psychological and pedagogical support of children in ordinary and special preschool institutions (Saint Petersburg and its region, Rostov-on-don and Murmansk). Diagnosis was carried out as a part of children’s routine examinations, after receiving written parental consent. The data was collected in 26 preschool institutions using the computer system “Longitude” (Software Longitude, Version 19, production of LLC “Longitude”, S-Petersburg, Russia) [14], as previously for elder children [13,15,16]. “Longitude” is regularly used in these institutions for annual surveys. The study used all the results of diagnostics of 3-year-old children, including children without a diagnosis (group Norm) and with an assessed diagnosis (groups DD/ASD), in the period from 2015 to 2019. The data was transmitted to us in an impersonal form.

“Longitude” software included a large bank of tasks, presented in accordance with the child’s calendar age, and aimed at estimating a wide range of abilities in primary domains such as motor skills, social adaptation, and cognitive abilities. A psychologist worked with the child, conducted the evaluations, and filled out the test’s electronic forms. To collect data in different conditions, offline and online versions of the program have been developed that allowed obtaining quickly representative data from different regions of Russia. Herewith, for all the participants the same standardized instructions and stimulus materials were used as a part of the software during the process of data collection. The software choice as the main form of implementation of the data collection was due to both the described data collection capabilities at the research stage, and the literature data on the higher efficiency of computer techniques in their application. The choice of data collection computerized form was due to the greater practicality and simplicity of implementation, but also, to the higher efficiency of application. For example, Carroll et al. showed that the use of computer-based cognitive development screenings for the purpose of detecting DD significantly increased the number of children diagnosed and those ones whose development was corrected timely at an early age [17].

The content of the questions and tasks was typical for screenings and development tests, but at the same time very versatile, since the bank of tasks was created as the result of a survey of a large number of expert practitioner psychologists. In this study, the “Longitude” software was used only as a tool to collect raw data for subsequent analysis, without taking into account the grouping of tasks in the structure of the original method; thus we worked with completely “clean” data, not distorted by the a priori subtest or factor structure of the method used.

The initial tasks bank included 502 tasks for children aged from 2 months to 7 years of which tasks relevant for 3-year-olds were selected. As a result of reducing the responses to tasks to a uniform dichotomous form, the initial data for analysis included 847 dichotomous items: 1-cannot perform, 2-can perform. The full content of items, tasks and stimuli materials is presented on the project’s website http://info11.testpsy.net/.

Statistical analysis was performed for the following purposes: (1) identification of scales which have factor validity and sufficiently high reliability for the “Norm” sample; (2) selection of scales that have the highest discriminative validity according to the “Norm–DD” criterion (diagnosis), validity in content; (3) interpretation of the relationships between scales-predictors, age and their relative contribution to the prediction of the diagnosis; and (4) standardization of the scales for 3-year-olds (the “Norm” sample). Statistical analysis was performed using IBM SPSS Statistics and AMOS 25 version (IBM Corp. Released 2017. IBM SPSS Statistics & AMOS for Windows, Version 25.0. Armonk, NY, USA).

## 3. Results

### 3.1. A Preliminary Selection of Items and the Scales Formation

A preliminary analysis of the diagnostic results showed that children with ASD for most of the tasks of the method obtained indeterminate results, differing from “can perform”-“cannot perform”. Therefore, data collected on a sample of children with ASD were used at this stage only for qualitative analysis and for tasks bank replenishment according to the next stages of the project, and thus were not used in statistical analysis.

233 items were selected out of 847 items used in the original methodology, for which responses to one of the two alternatives for this sample were no more than 95%. Of 233 binary items, 155 items were selected, which meet two requirements: (a) differentiating Norm and DD sample (by discrimination rate); (b) positively correlating with the age (in days). Further, the factor analysis of 155 variables was conducted on the sample Norm (*n* = 814) by the principal component method with varimax rotation to get the following results: (a) each variable has a factor loading not lower 0.40, with only one factor; (b) each factor forms a fairly reliable scale (Cronbach’s alpha not lower, than 0.70). Thus, 27 items were selected, and the distribution of the correct answers to each of them is presented in Table 1.

The factors are named in accordance with the items included in them (see Table 2): “Motor skills” (S1)’, performing all the 9 items which requires that the child has developed motor skills, realized with direct visual control; “General awareness” (S2), as all 6 items included in this factor, are connected with the breadth of the child’s knowledge about the world; “Logical reasoning” (S3), as the most tasks included in this factor require the child’s reasoning and the certain level of understanding; and “Executive functions” (S4), because this factor includes the tasks related to voluntary attention, short-term (operative) visual memory, primary counting skills, verbal-conceptual thinking.

### 3.2. Verification of Factor Validity and Reliability of the Scales

For verification of the factor validity of the 4-scale method, the confirmatory factor analysis (CFA) was implied. One-factor model (Model 1), 4-factor model with correlating factors (Model 2), hierarchical 4-factor model with a secondary factor (Model 3) and hierarchical model with the added connections between residues (Model 4) were compared. Goodness of fit indices for the models are presented in Table 3.

The models were compared by statistical significance of the difference of Chi-square values for the corresponding difference in degrees of freedom. Model 2 fits better to the sample data than Model 1 (*p* < 0.001), and Model 3 fits better to the sample data than Model 2 (*p* < 0.05). So, the best is the 4-facor model with secondary factor. However, according to the fit indices, Model 3 does not fit well enough to the sample data. That is why in Model 4, 6 correlations between the remnants errors of explicit variables (items) were added. Taking into consideration the high value of DF and the large number of the sample (814), 12 times exceeding the number of estimated parameters (64), the fit indices of the model indicate sufficient fit with the sample data (CFI, GFI > 0.9; RMSEA < 0.05) [19,20]. Thus, the factor validity of the 4-scale method was confirmed: (a) each of 27 items is determined statistically significant by only one factor (*p* < 0.001); (b) sets of indicators for each factor ensure the statistical significance of its variance (*p* < 0.001).

Each factor includes a set of items (tasks), ensuring the sufficient reliability of the corresponding scale Cronbach’s alpha at about 0.75 and more (Table 2). For the total scale of 27 items, Cronbach’s alpha is still higher and equals 0.874. Further, the scales values were calculated for each child as the sum of items, included in the corresponding factor (S1–S4); SS–is the sum of all the 27 items.

### 3.3. Verification of the Discriminative Validity of the Scales

For verification of discriminative validity of the scales, their mean values for the samples Norm and DD were compared. Additionally, the effect size of Cohen’s d was calculated. The results are given in Table 4.

Differences in each scale can be considered very sufficient (d >> 0.80). Most of the differences are in “Logical reasoning” scale (S3), the least are in the “General awareness” scale (S2). Convincity of evidence was obtained in favour of discriminative validity of the scales according to the criterion for differentiating groups Norm/DD.

### 3.4. The Structure of Interconnections of the Scales with the Age

In Table 5, Pearson correlations of the scales between themselves and the age (in days) are given.

Structural equation modeling (SEM) was used to clarify the relationship structure. The hypothesis, that age directly affects each scale, was tested, and the scales are the indicators of the general factor G (Figure 1). Since the requirement of multidimensional normality is satisfied (Multidimensional Kurtosis −0.093; C.R. = 0.159), the maximum likelihood method was applied. The final model on the goodness of fit indices well corresponds to the sample data. All the estimated parameters (regression coefficients, covariance, variances) are statistically significant (*p* < 0.001). Thus, the original hypothesis was confirmed.

The structural model (Figure 1) shows that different components of abilities of 3-year-old children are differently affected by age and G-factor. So, the strongest effect the age has is on the “General awareness” (S2), while G-factor has the least influence on “General awareness” (S2). On the other hand, the most contribution of the G-factor is surveyed on “Logical reasoning” (S3), but the influence of age on “Logical reasoning” (S3) is minimal. In regards to “Motor skills” (S1) and “Executive function” (S4), the contribution of age and G-factor is almost the same.

### 3.5. Scales Adjusting According to the Age

At the previous stage, the undeniable influence of age on the 4 scale values was revealed. Curve fit test showed that all connections are approximated only in a straight line (β-coefficients for 2 and 3 degrees are not statistically significant). That is why the age was recorded as before [13,15] by introducing a linear correction, eliminating the slope of the regression line.
Correction formula: Scor_i_ = S_i_ − eS_i_ + M_s_ = S_i_ − (b_0_ + b_1_Age_i_) + M_s_
where S_i_—is the initial value of the scale for a child I; Age_ii_—the age of the child (in days) I; eS_i_—esteem S_i_ by the equation of regression eS_i_ = b_0_ + b_1_A_i_; b_0_, b_1_—constant and regression coefficient; Ms—the mean value of the scale S.

After introducing such correction, the scale values increase (for “Junior”) or decrease (for “Senior”), and the influence of age levels off. Corrections for age in days were used for each of the 4 scales and for the total scale. For example, the equation for the total scale is as follows:SScor_i_ = SS_i_ − 0.53224 − Age_i_ × 0.0336 + 43.4742
where SS_i_—the initial value of the total scale for the child with the number i, and Age_i_—his/her age (in days).

### 3.6. Test Norms Elaboration

The test norms were elaborated for the total scale and separately for each of the four scales included in it. Standardization sample is 3-year-old children without the diagnosis (sample “Norm”, N = 814). Nonlinear standardization on algorithm is applied in a 10-item scale of “stens” [21], using percentile boundaries for scores that correspond to the normal distribution. This ensures that the distribution of standard scores on all scales for the “Norm” sample is normal. The results (test norms) for the Total scale (sample “Norm”) are presented in Table 6. In the last 2 lines of the test norm tables, the percentile of the sample “Norm” (% Norm) and DD (% DD) are presented for each value of the sten.

In terms of optimal limit to predict belonging to DD group, the sten limit 2 and 3 (“not higher than sten 2”) can be considered. In such cases, we can predict the DD forecast accuracy at 92.3% (24 of 26 children of DD group were predicted correctly), and the specificity of the “Norm” forecast at 93.4% (54 of 814 children of “Norm” group were assigned to DD group). Thus, 6.6% of children of the “Norm” group represent a risk group and need more detailed diagnosis by experts. For more detailed characteristics of the effectiveness of the test norms, see Table 7.

## 4. Discussion

This work became the next stage of our long-term research based on psychometric approach in elaboration screening tools. When developing this screening, we strictly adhered to the current recommendations for the psychometric tools standards in the test construction concerning such properties as reliability and specificity [22]. Looking forward, procedures of the screening standardization and accuracy computations will be our next steps. In order to avoid common biases in screening test elaboration [23], we carried out our research on a huge sample, used initially wide range of test tasks to face possible non-equivalent test bias and the process of data collection was strictly identical for each participant to prevent a procedural bias. However, the weak point of our research is a relatively small number of children assessed with a diagnosis (DD/ASD), understood as a spectrum bias, which is reflected in the limitations section and which nonetheless can be partially overcome by the large sample size and high reliability and specificity of the forecast. Discussing the results, first of all, it should be emphasized that the elaborated screening scales can only be used for early diagnosis of children with DD without autistic-like symptoms presence after required standardization and validation procedures. Tasks in the scales assume the child’s ability to social interaction at the level of instruction understanding and its implementation. For children with autistic-like symptoms, first comes a lack of social interaction [24], limitations in the instructions understanding [25], repetitive behaviours with elements of self-stimulation [26], and stimulus overselectivity in relation to relevant stimulus [27]. In addition, these children often show a refusal to cooperate in a new environment with new people [28]. For early detection of autistic-like developmental disorders, it is necessary to evaluate the child’s behavioral and cognitive skills using parental questionnaires and check-lists of spontaneous behavioural activity of the child.

It is important to note that the identified structure of scales for 3-year-olds is similar to the previously identified scales sets for 4- and 5-year-olds [13], according to the factors of logical reasoning, motor development and general awareness, but differs from the previously identified scales sets for 6-year-olds, where the most important factors in predicting the diagnosis were sustained attention and counting [29]. For 3-year-olds, the executive functions factor is additionally allocated. At the same time, the obtained models correspond to the Denver screening [9] for motor development factor; speech development and communication presented in the Western screening in our model correspond to the General awareness factor, which assumes speech development of the child sufficient enough to expand his/her world comprehension and form communication at a level that allows to accumulate verbal knowledge; social adaptation in our models is not represented. However, for each age, the factor of Logical reasoning was additionally identified, and for 3-year-olds, the factor of Executive functions, which in fact reflects the structural and functional maturation of the brain and can be considered as the necessary neurophysiological basis for adaptive behavior. The weak negative link between errors (residuals) in the model (see Figure 1) connecting “General Awareness” and “Executive functions” can be explained as follows: when a child has a delay in a maturation of the frontal cortex, which is reflected in an underdevelopment of executive functions [30,31], the main compensatory mechanism for children with DD is the mastering of verbal abilities and first of all an expansion of general awareness [32].

The differences between typically developing children and children with a DD for 3-year-olds on each of the obtained scales are very significant (Cohen’s d >> 0.80). Thus, there is strong evidence in favor of the scales discriminative validity based on the criterion of differentiation of Norm/DD groups. The largest differences were found on the scales “Logical reasoning” and “Motor skills”, the smallest–on the scale “General awareness”. It is important to note that similar results were obtained for 4- and 5-year-olds: the greatest contribution to the differentiation of the groups Norm/DD for each age was implemented by the Logical reasoning factor, the smallest although significant, by General awareness factor [13]. This result was quite expected, as it is known that the ability to reason logically belongs to the structure of nonverbal intelligence, reflects general intellectual abilities, and is largely due to biological inclinations and heredity, and, accordingly, is subject to training to a much lesser extent than verbal abilities. On the other hand, the smallest differences between groups for each age were obtained on the “General awareness” scale, which is also quite natural. The process of forming and expanding the child’s awareness about the world primarily reflects the time spent on classes with the child and is in fact the process of an information accumulation. This process can occur without additional comprehension or logical processing if they are beyond the child’s present capabilities, but if there are sufficient resources for memorization, often mechanical ones.

Structural equation modeling (Figure 1) of the structure of the scales relationship with the age shows that the 4 selected scales are indicators of the General intelligence factor (G). Likewise for 4–5-year-olds, the age does not inflict a direct effect on General intelligence, but affects its indicators (scales) in different ways. It was found that the age has the greatest influence on “General awareness” (S2), while G factor has the least influence on “General awareness” (S2). “General awareness” traditionally refers to verbal abilities, while it is obvious that the older the child becomes, the wider his/her knowledge of the world, so the age naturally affects the development of General awareness. However, G factor, responsible for the success of intellectual tasks in general, affected General awareness to the least extent. This is probably due to the fact that General awareness is subject to training to the greatest extent and cannot serve as a reflection of a child’s general intellectual abilities. At the same time, the greatest contribution of factor G is observed in “Logical reasoning” (S3), an ability primarily related to non-verbal intelligence, reflecting the child’s highly inherited abilities, which are subject to training to a much lesser extent than verbal abilities. Interestingly, the influence of the age on “Logical reasoning” (S3) for 3-year-olds is minimal, which supports nativist theories that assume that a child has a certain set of innate abilities, the development of which is due to the deployment of genetic programs [33] and by this age this ability is already represented in the structure of intelligence.

The fact that the scale values significantly correlate with the age in days was taken into account when developing test norms by introducing an adjustment for the child’s age in days using the linear regression model. As a result, for the same number of correctly completed tasks, the scale values for younger 3-year-old children are higher than those for older ones. The developed test norms demonstrate a high accuracy of more than 90% for distinguishing the Norm and DD groups.

## 5. Conclusions

The main result of our current research is the elaboration of a high-precise (at least 90%) rapid DD diagnosis in 3-year-olds, which allows identifying quickly the “risk group” for further clarification of the diagnosis and an educational route calculation. The expected accuracy of the prediction when using the elaborated screening is significantly higher than that in known sources (more than 90%). For example, Dawson [9], in the article on testing Denver screening in clinical practice, found a prediction accuracy of less than 70% sufficient. It was found out that the scales for 3-year-olds, as the areas in which the Norm and DD samples differ most strongly, are broadly similar to the scales obtained for older preschoolers, and the most significant differences are shown in the scales “Logical reasoning” and “Motor skills”.

The study also showed that the initially wide range of used tasks is not adequate for diagnosing children with ASD. As was shown by the preliminary analysis of the data, the tasks suitable for children with DD gave an uncertain result for the children with ASD. Thus, for the diagnosis of ASD, more specific features should be considered such as the characteristics of behavior, social interaction and perception, information about which can be obtained from a survey of the child’s immediate environment. Thereby, the development of methods for rapid diagnostics of children with ASD is our next immediate task. The applied result of solving this problem should be creation of an online screening system for the diagnosis and differentiation of these developmental problems, developed on the basis of the current version of the research software and available to specialists of educational and medical institutions.

## 6. Limitations

Our study is limited by the fact that the sample of children with assessed diagnosis was noticeably smaller than the one of children belonging to the norm group. This is due to the current situation of diagnostics in Russia, when for the first time children are diagnosed for deviations in mental development only at the age of 3 years and later on, when they enter pre-school educational institutions. Obtained parameters of the screening sensitivity and specificity are preliminary. Further research is needed to clarify the quantitative parameters of the developed screening accuracy. In this paper, we focused on the implementation of the psychometric approach in the screening elaboration. The next stage of the study will be devoted to the testing of the developed screening.

## Figures and Tables

**Figure 1 diagnostics-10-00646-f001:**
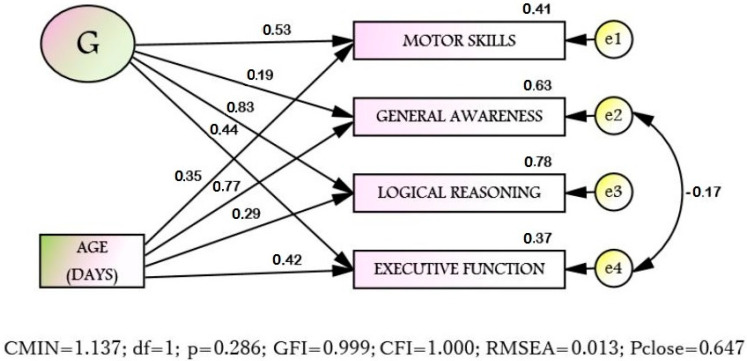
The structural model of cognitive and psychomotor development of 3-year-old children in Russia (*n* = 814).

**Table 1 diagnostics-10-00646-t001:** Distribution of correct answers to 27 items for the Norm and DD samples.

Item	Norm (*n* = 814)		DD (*n* = 26)		Total (*n* = 840)	
	Count	%	Count	%	Count	%
n182	719	88.33	5	19.23	724	86.19
n187	525	64.50	4	15.38	529	62.98
n189	662	81.33	1	3.85	663	78.93
n195	640	78.62	4	15.38	644	76.67
n238	685	84.15	5	19.23	690	82.14
n248	504	61.92	2	7.69	506	60.24
n260	464	57.00	2	7.69	466	55.48
n266	687	84.40	5	19.23	692	82.38
n270	553	67.94	1	3.85	554	65.95
n288	394	48.40	2	7.69	396	47.14
n289	597	73.34	0	0.00	597	71.07
n291	556	68.30	3	11.54	559	66.55
n295	476	58.48	4	15.38	480	57.14
n302	323	39.68	1	3.85	324	38.57
n325	404	49.63	1	3.85	405	48.21
n355	448	55.04	1	3.85	449	53.45
n361	588	72.24	4	15.38	592	70.48
n362	605	74.32	5	19.23	610	72.62
n385	266	32.68	1	3.85	267	31.79
n548	748	91.89	8	30.77	756	90.00
n628	407	50.00	4	15.38	411	48.93
n629	293	36.00	2	7.69	295	35.12
n640	320	39.31	7	26.92	327	38.93
n655	194	23.83	3	11.54	197	23.45
n818	564	69.29	3	11.54	567	67.50
n847	631	77.52	4	15.38	635	75.60
n890	157	19.29	0	0.00	157	18.69
Mean	497	61	3	12	500	59

Since items distributions are mostly nonsymmetrical and the data is binary, Categorical Principal Components Analysis (CATPCA) was applied to them: Scaling level-ordinal, Normalization method–variable principal, Rotation method–quartimax with Kaiser normalization [18]. The main results of this stage of analysis are presented in Table 2, containing tasks, grouped into scales (S1–S4). DD: developmental delay.

**Table 2 diagnostics-10-00646-t002:** Main results of Categorical Principal Components Analysis and reliability check on Cronbach’s alpha scales.

Rotated Component Loadings of 27 Items (*n* = 814), α = 0.874, 43.92% of Variance	CL ^1^
Factor 1: “Motor skills” (S1; 13.48% of variance), *α* = 0.755 (9 items)	
195. Draws a cross by himself looking at the example	0.710
248. Draws a cross without looking at the example	0.640
260. Draws a person (“head and legs”).	0.591
325. Can cut with scissors in a straight line	0.571
187. Buttons buttons	0.553
238. Can complete the picture of a person with missing arms and legs	0.553
355. Walks “heel to toes”	0.515
302. Can copy a triangle	0.502
266. Can hold back on signal	0.413
Factor 2: “General awareness” (S2; 11.00% of variance), *α* = 0.792 (6 items)	
640. (639–640) The child knows the names of wild animals ^2^	0.777
628. The child gives a correct answer to the question: “How many ears have you got?”	0.746
655. (654–655) The child knows the names of cubs of some animals ^2^	0.717
385. Can find true and false in the picture.	0.673
890. The child gives the correct answer to the question: “What is left on the ground after raining?”	0.545
629. The child knows all seasons	0.506
Factor 3: “Logical reasoning” (S3; 10.54% of variance), *α* = 0.745 (7 items)	
291. Counts the fingers correctly	0.662
289. The child gives the correct answers to the questions: “When do you go to bed, in the morning or in the evening? When do you have lunch, at night or in the afternoon? When do you sleep?”	0.640
189. Can answer the questions, like: “What do you do, if you feel cold? when you are tired? when you are hungry?”	0.585
270. Knows six colours	0.579
182. Says his first name and his surname	0.480
295. The child uses grammatically correct sentences in his speech	0.471
288. “The child gives the correct answers to the question: “What season is it now? During which season it can snow/During which season the leaves get yellow and fall?”	0.453
Factor 4: “Executive functions” (S4; 8.90% of variance), *α* = 0.759 (5 items)	
548. (547–548) The child finds 10 pictures of 20, which you showed to him before ^2^	0.667
847. (846–847) The child can answer the questions: “What do you think, what is in common between a spoon and a fork?” ^2^	0.667
362. Can count up to 5 objects	0.661
818. (816–818) Counts the objects correctly. ^2^	0.658
361. Finds a picture in a row, which differs from the rest	0.560

^1^—component loadings; ^2^—in cases, where only one sub-item was taken–that is, a certain level of performing non-dichotomy item on the whole, the number of this item is given in brackets. S1: Motor skills; S2: General awareness; S3: Logical reasoning; S4: Executive functions.

**Table 3 diagnostics-10-00646-t003:** Goodness of fit indices for the models (Confirmatory Factor Analysis).

Model	CMIN ^1^	*df*^2^; *p*^3^	CFI ^4^	GFI ^5^	RMSEA ^6^	90% CI ^7^
1	2783.405	324; *p* < 0.001	0.588	0.763	0.097	0.093–0.100
2	1368.936	318; *p* < 0.001	0.824	0.888	0.064	0.060–0.067
3	1375.091	320; *p* < 0.001	0.823	0.887	0.064	0.060–0.067
4	883.726	314; *p* < 0.001	0.904	0.926	0.047	0.044–0.051

CMIN ^1^—Chi-square, df ^2^–degrees of freedom, *p*
^3^—*p*-level of significance, CFI ^4^—comparative fit indices, GFI ^5^—goodness-of-fit statistic, RMSEA ^6^—root mean square error of approximation, 90% CI ^7^—limits of confidence interval for RMSEA [19,20].

**Table 4 diagnostics-10-00646-t004:** Comparison of scales means for samples Norm (*n* = 814) and DD (*n* = 26) ^1^.

Scales	Diagnosis	Means	Std. Dev.	Cohen’s d
Motor skills (S1)	Norm	14.7494	2.39127	2.394
	DD	9.9615	1.50946	
General awareness (S2)	Norm	8.0111	1.93781	0.842
	DD	6.6538	1.19808	
Logical reasoning (S3)	Norm	11.8612	1.95248	2.795
	DD	7.6154	0.89786	
Executive functions (S4)	Norm	8.8526	1.47480	1.956
	DD	5.9231	1.52113	
Sum (SS)	Norm	43.4742	5.84327	2.709
	DD	30.1538	3.77033	

^1^ The statistical significance of the difference on *t*-Student tests on all the scales is *p* < 0.001 (with the Benjamini-Hochberg correction for multiple comparisons).

**Table 5 diagnostics-10-00646-t005:** Pearson correlations of scales and age (*n* = 814) ^1^.

	Age	S1	S2	S3	S4	SS
**Age (days)**	1	0.352	0.771	0.285	0.420	0.601
**Motor skills (S1)**	0.352	1	0.358	0.543	0.392	0.808
**Awareness (S2)**	0.771	0.358	1	0.382	0.327	0.688
**Logical reasoning (S3)**	0.285	0.543	0.382	1	0.481	0.805
**Executive functions (S4)**	0.420	0.392	0.327	0.481	1	0.682
**Sum (SS)**	0.601	0.808	0.688	0.805	0.682	1

^1^ all correlations are significant at *p* < 0.01 (with the Benjamini-Hochberg correction for multiple tests).

**Table 6 diagnostics-10-00646-t006:** Test norms for the total development scale for the sample of 3-year-old children of the “Norm” sample and percent for the stens in sample “Norm” and “DD”.

SScor ^1^ (Upper Limit)	30.19	32.83	35.8	38.57	41.46	44.05	46.26	47.95	49.37	50.88	>50.9
**Sten**	0	1	2	3	4	5	6	7	8	9	10
**Norm, %**	0.6	1.7	4.3	9.2	15.0	19.2	19.2	15.0	9.2	4.3	2.3
**DD, %**	61.5	23.1	7.7	3.8	3.8	0.0	0.0	0.0	0.0	0.0	0.0

^1^ raw score, the total value on the total scale, adjusted for the child’s age (in days) according to the regression model.

**Table 7 diagnostics-10-00646-t007:** Indicators of test norms effectiveness.

Statistic	Value	95% CI
Sensitivity	92.31%	74.87–99.05
Specificity	93.37%	91.43–94.98
Positive Predict Value	99.62%	99.50–99.72
Accuracy	92.36%	90.35–94.07

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
