# Peer review of "Elaboration of Screening Scales for Mental Development Problems Detection in Russian Preschool Children: Psychometric Approach"

_diagnostics, 2020, doi:10.3390/diagnostics10090646_

Round 1

Reviewer 1 Report

Abstract:

Please provide more quantitative indices in addition to the forecast accuracy.

Introduction:

How could the authors provide justifications about the reliability of using the 3-year-old indices? Please discuss.

Methods:

How was the sample size estimated in this study?

What is the possible bias?

What are the inclusion and exclusion criteria?

The reviewer invites the authors to provide the items of the survey as supplementary.

Results:

Please provide the descriptive statistics table indicating the samples analyzed in the study in different sub-categories.

When reporting the discriminative capability of the method, training, and test sets must be used, and the performance indices of the test sets, including sensitivity, specificity, PPV, and accuracy, are reported. Moreover, based on the STARD guideline, the CI 95% of the performance indices must be reported.

Discussion:

The authors must discuss the limitations of the study, including possible bias and future directions.

Any method as the state-of-the-art for comparison?

Author Response

Dear Reviewer, thank you very much for your attention to our work ans useful recommendations. Our answers you will find in the uploaded file. Kind regards

Answers to the Reviewer 1

Deeply honored Reviewer,

First of all let us express our sincere gratitude for your precise attention to our work and your useful comments and suggestions.

  • R Abstract: Please provide more quantitative indices in addition to the forecast accuracy.

Answer: The indicators of sensitivity (92.3%) and specificity (93.4%) obtained in this study are preliminary, since the sample of 3-year-old children with DD diagnosis is still small. More definite quantitative characteristics of forecast accuracy will be obtained after the approbation and wider usage of the screening.

  • R Introduction: How could the authors provide justifications about the reliability of using the 3-year-old indices? Please discuss.

Answer: It is not clear what we are asked about. If by indices you mean developed scales, then their reliability in terms of items internal consistency is justified using the Cronbach's alpha.

  • R Methods:
  1. a) How was the sample size estimated in this study?

Answer: The data was collected in 26 preschool institutions using the computer system “Longitude”, which is regularly used in these institutions for annual surveys. The study used all the results of diagnostics of 3-year-old children, including children without a diagnosis (group Norm) and with an assessed diagnosis (groups DD/ASD), in the period from 2015 to 2019.

We made changes in the manuscript on p. 3, the lines 98-103 as a response on this question.

  1. b) What is the possible bias?

Answer: We addressed the issue in the discussion section (see p. 10, the lines 283-289). Also, we added the limitations section where we addressed possible bias of our research and future directions. Respectively the changes were made on p.12, lines 378-386.

  1. c) What are the inclusion and exclusion criteria?

Answer: The results of the routine examinations of all children for the entire specified period (2015-2019) were obtained. No results were excluded from the analysis.

  1. d) The reviewer invites the authors to provide the items of the survey as supplementary.

Answer: The table on the distribution of the correct responses to 27 items for the Norm and DD samples was added in the manuscript on p. 4, the line 152 and also is presented below:

Item

Norm (N=814)

DD (N=26)

Total (N=840)

Count

%

Count

%

Count

%

n182

719

88.33

5

19.23

724

86.19

n187

525

64.50

4

15.38

529

62.98

n189

662

81.33

1

3.85

663

78.93

n195

640

78.62

4

15.38

644

76.67

n238

685

84.15

5

19.23

690

82.14

n248

504

61.92

2

7.69

506

60.24

n260

464

57.00

2

7.69

466

55.48

n266

687

84.40

5

19.23

692

82.38

n270

553

67.94

1

3.85

554

65.95

n288

394

48.40

2

7.69

396

47.14

n289

597

73.34

0

0.00

597

71.07

n291

556

68.30

3

11.54

559

66.55

n295

476

58.48

4

15.38

480

57.14

n302

323

39.68

1

3.85

324

38.57

n325

404

49.63

1

3.85

405

48.21

n355

448

55.04

1

3.85

449

53.45

n361

588

72.24

4

15.38

592

70.48

n362

605

74.32

5

19.23

610

72.62

n385

266

32.68

1

3.85

267

31.79

n548

748

91.89

8

30.77

756

90.00

n628

407

50.00

4

15.38

411

48.93

n629

293

36.00

2

7.69

295

35.12

n640

320

39.31

7

26.92

327

38.93

n655

194

23.83

3

11.54

197

23.45

n818

564

69.29

3

11.54

567

67.50

n847

631

77.52

4

15.38

635

75.60

n890

157

19.29

0

0.00

157

18.69

Also we give the link to the full content of items, tasks and stimuli materials which are available on the project's website http://info11.testpsy.net/ (see footnotes below Table 2, p. 6, the lines 162-163).

  • R Results:

  1. Please provide the descriptive statistics table indicating the samples analyzed in the study in different sub-categories.

Answer: Table 4 shows descriptive statistics and the number of samples surveyed (see p. 7, the line 207). Or is this question addressed to descriptive statistics for the original 27 items? The table of distribution of correct answers is given above and in the text of the article (p. 4, the line 152).

  1. When reporting the discriminative capability of the method, training, and test sets must be used, and the performance indices of the test sets, including sensitivity, specificity, PPV, and accuracy, are reported. Moreover, based on the STARD guideline, the CI 95% of the performance indices must be reported.

Answer: Following your recommendation we added Table 7 “Indicators of test norms effectiveness” (see p. 10, the line 277). Performance indices will be available as the data on 3-year-olds who have already been diagnosed with DD is accumulated. In this article we have focused on the implementation of the psychometric approach in the development of screening. As we point out in the discussion (p. 10, the lines 279-283) and in the limitation section (p. 12, the lines 378-386).

  • R Discussion:

  1. The authors must discuss the limitations of the study, including possible bias and future directions.

Answer: Thanks for the recommendation. The reference to the possible study bias was added in the discussion section (p. 10, the lines 283-289). Also, the limitations section was added (p. 12, the lines 378-386).

  1. Any method as the state-of-the-art for comparison?

Answer: Surprisingly and unfortunately, we searched for modern screenings far and wide but couldn’t find anything new except for long existing ones, such as Denver, Griffiths, Vineland. We quoted Denver in our article. Would you recommend us to quote some of the others?

Additional: In order to improve English language and style we will apply to the MDPI English editing service as soon as there no more correction needed of the manuscript’s content.

Reviewer 2 Report

My review will be based on the statistical methodology that was followed as this is the point of major consideration. While the authors present an analysis conducted skillfully, I am afraid the choice of methods it is not supported by current literature. 

To begin with, SPSS and AMOS have been used, meaning that the authors consider the data approximately normally distributed. Is that the case? IS the response scale five or more points on ordinal scale? Are the responses symmetrical? This is not clear in the manuscript. Otherwise please consider running a categorical factor analysis in MPlus, Stata or R.

Assuming that the common factor model is appropriate, the principal components method for factor extraction is definitely not. The method is not used anymore in the context of factor analysis due to its lack of acknowledging measurement error. The authors need to choose a better method based on the distribution of the data and on the existence of missing values. Also, why varimax rotation? It is not clear to me why one expects the different dimensions not to be related?

To the major commonly made  EFA mistakes, I would also like to draw attention to the CFA correlated error terms to ensure identification of the model. Unless if this correlation was a-priori expected or can be properly and clearly interpreted, adding correlated error terms is not recommended as it is data driven in a theory driven model, and it only shows a bad model. 

My advice to the authors is to seek advise for the analysis from a specialist psychometrician. 

Author Response

Dear Reviewer, thank you very much for your attention to our work and useful recommendations. Our answers you will find in the uploaded file. Kind regards

Answers to the Reviewer 2

Deeply honored Reviewer,

First of all let us express our sincere gratitude for your precise attention to our work and your useful comments and suggestions.

  • R: To begin with, SPSS and AMOS have been used, meaning that the authors consider the data approximately normally distributed. Is that the case?

Answer: SPPS and AMOS allow you to work with the data which is not distributed normally due to the availability of the options that enable you to work with such data. For example categorical principal components analysis (CATPCA) or Asymptotically Distribution Free (Method of Estimate) in AMOS.

  • R: Is the response scale five or more points on ordinal scale?

Answer: Thanks for your question. We clarified that all the items are binary (see p.3, the line 144).

  • R: Are the responses symmetrical?

Answer: No, but the asymmetry is not so great (table 1 was included in the article on p. 4, the line 152) to assume that parametric factor analysis gives significant errors (see Kline R. 2011, p. 63). ). N548 has the largest asymmetry (skew=3.069), while the others have significantly less. However, taking into account your recommendations, we applied categorical principal components analysis.

  • R: Otherwise please consider running a categorical factor analysis.

Answer: Thank you for the recommendation. We applied a categorical factor analysis and got not identical, but very similar results. The results of the analysis are included in the article on pp. 4-5, the lines 153-163.

R: Assuming that the common factor model is appropriate, the principal components method for factor extraction is definitely not. The method is not used anymore in the context of factor analysis due to its lack of acknowledging measurement error.

Answer: Discussions about the applicability of PCA as one of the methods of factor analysis are still ongoing (Thompson, 2004, Fabrigar, Wegener, MacCallum, & Strahan, 1999, Mulaik, 1992). Yes, of course, such FA methods as the Maximum Likelihood Method are much more theoretically justified, since they are aimed at the most accurate reproduction of the original correlations. But, taking into account the huge number of sources of errors in psychological measurements, the use of a simpler PCA is justified. In addition it can be added that when the number of measured variables is large - also with respect to the supposed number of latent variables (as is the case in our study) PCA and FA lead to the similar results (Thompson, 2004; Fabrigar et al., 1999).

  • R: The authors need to choose a better method based on the distribution of the data and on the existence of missing values.

Answer: First of all, we don't have missing values, since each item is binary and the response can be either the child can perform the task or he/she cannot perform the task. Given the questionable symmetry of the distributions, we used your recommendation and applied CATPCA. The results are shown in Table 2 (on pp.  4-5, the lines 153-163), and although they differ from the previous ones, the difference is insignificant. Probably this is due to the large sample size. You will also find full printouts of the SYNTAX and OUTPUT: CATPCA at the end of our responses.

  • R: Also, why varimax rotation? It is not clear to me why one expects the different dimensions not to be related?

Answer: Of course, the scales correlate, as indicated in Table 5 on p. 8, the line 219, and on Figure 1, p. 9. Nevertheless, We prefer orthogonal rotation options, as they have a more transparent interpretation. In addition, psychometrics has empirically developed the rule of "factor loading not less than 0.4" for including item into factor to obtain a reliable scale, after an orthogonal rotation (Kline, 2000; Cooper, 2010). In our case, the most optimal method was the Quartimax rotation with Kaiser Normalization (see Table 2 on pp. 4-5, the lines 153-163; also see full printouts at the end of our responses in this file).

  • R: To the major commonly made  EFA mistakes, I would also like to draw attention to the CFA correlated error terms to ensure identification of the model. Unless if this correlation was a-priori expected or can be properly and clearly interpreted, adding correlated error terms is not recommended as it is data driven in a theory driven model, and it only shows a bad model. 

Answer: Of course, such surprises require either a priori assumptions or distinct a posteriori interpretations. We used CFA primarily to show that the 4-factor model is better than the 1-factor model. Regarding the assumption of correlations between errors for Model 4 there are many situations—particularly with respect to social psychological research where these parameters can make strong substantive sense and therefore should be included in the model (Joreskog, & Sorbom, 1993). In our case, correlations between errors are explained by the similarity of the corresponding task pairs. Correlate errors for tasks that are similar in one way or another: n629 – n288 (knows the seasons), n195 – n248 (draws a cross), n291 – n362 (can count), and so on.

If you are referring to the model shown in Figure 1, on p. 9, where 1 link is added between 2 errors, then we consider this acceptable if there are not very many such links. We added the interpretation in the discussion section on the p.11, the lines 313-317 as following: The weak link between errors (residuals) can be explained by the fact that when a child has a delay in a maturation of the frontal cortex, which is reflected in an underdevelopment of executive functions, the main compensatory mechanism for children with DD is the mastering of verbal abilities and first of all an expansion of general awareness.

Additional: In order to improve English language and style we will apply to the MDPI English editing service as soon as there no more correction needed of the manuscript’s content.

SYNTAX:

CATPCA VARIABLES=n187 n189 n238 n248 n260 n266 n270 n288 n289 n291 n295 n302 n325 n355 n361 n362

    n385 n548 n628 n629 n640 n655 n847 n890 n818 n195 n182

  /ANALYSIS=n187(WEIGHT=1,LEVEL=ORDI) n189(WEIGHT=1,LEVEL=ORDI) n238(WEIGHT=1,LEVEL=ORDI)

    n248(WEIGHT=1,LEVEL=ORDI) n260(WEIGHT=1,LEVEL=ORDI) n266(WEIGHT=1,LEVEL=ORDI)

    n270(WEIGHT=1,LEVEL=ORDI) n288(WEIGHT=1,LEVEL=ORDI) n289(WEIGHT=1,LEVEL=ORDI)

    n291(WEIGHT=1,LEVEL=ORDI) n295(WEIGHT=1,LEVEL=ORDI) n302(WEIGHT=1,LEVEL=ORDI)

    n325(WEIGHT=1,LEVEL=ORDI) n355(WEIGHT=1,LEVEL=ORDI) n361(WEIGHT=1,LEVEL=ORDI)

    n362(WEIGHT=1,LEVEL=ORDI) n385(WEIGHT=1,LEVEL=ORDI) n548(WEIGHT=1,LEVEL=ORDI)

    n628(WEIGHT=1,LEVEL=ORDI) n629(WEIGHT=1,LEVEL=ORDI) n640(WEIGHT=1,LEVEL=ORDI)

    n655(WEIGHT=1,LEVEL=ORDI) n847(WEIGHT=1,LEVEL=ORDI) n890(WEIGHT=1,LEVEL=ORDI)

    n818(WEIGHT=1,LEVEL=ORDI) n195(WEIGHT=1,LEVEL=ORDI) n182(WEIGHT=1,LEVEL=ORDI)

  /DIMENSION=4

  /NORMALIZATION=VPRINCIPAL

  /MAXITER=100

  /CRITITER=.00001

  /ROTATION=QUARTIMAX KAISER

  /RESAMPLE=NONE

  /PRINT=LOADING(SORT)

  /PLOT=OBJECT(20).

CATPCA - Principal Components Analysis for Categorical Data

Credit

CATPCA

Version 2.0

by

Leiden SPSS Group

Leiden University

The Netherlands

Case Processing Summary

Valid Active Cases

840

Active Cases with Missing Values

0

Supplementary Cases

0

Total

840

Cases Used in Analysis

840

Iteration History

Iteration Number

Variance Accounted For

Loss

Total

Increase

Total

Centroid Coordinates

Restriction of Centroid to Vector Coordinates

0a

12,405885

,000040

95,594115

95,594115

,000000

1b

12,405885

,000000

95,594115

95,594115

,000000

a. Iteration 0 displays the statistics of the solution with all variables, except variables with optimal scaling level Multiple Nominal, treated as numerical.

b. The iteration process stopped because the convergence test value was reached.

Model Summary

Dimension

Cronbach's Alpha

Variance Accounted For

Total (Eigenvalue)

% of Variance

1

,893

7,155

26,500

2

,572

2,225

8,241

3

,439

1,733

6,419

4

,235

1,293

4,788

Total

,955a

12,406

45,948

a. Total Cronbach's Alpha is based on the total Eigenvalue.

Model Summary Rotationa

Dimension

Cronbach's Alpha

Variance Accounted For

Total (Eigenvalue)

% of Variance

1

,867

4,285

15,870

2

,783

2,993

11,086

3

,824

2,659

9,848

4

,779

2,469

9,143

Total

,955b

12,406

45,948

a. Rotation Method: Quartimax with Kaiser Normalization.

b. Total Cronbach's Alpha is based on the total Eigenvalue.

Component Loadings

Component Loadings

Dimension

1

2

3

4

n361

,628

-,049

,256

-,268

n291

,608

-,171

,206

,362

n189

,607

-,205

-,037

,256

n362

,602

-,202

,493

,009

n818

,587

-,138

,475

-,055

n289

,562

-,162

-,003

,380

n629

,558

,331

,124

,008

n248

,551

-,194

-,310

-,124

n295

,546

-,014

-,203

,270

n195

,533

-,258

-,334

-,246

n260

,529

-,071

-,290

-,172

n288

,520

-,073

,021

,224

n270

,519

-,111

-,071

,392

n325

,512

-,102

-,213

-,235

n238

,508

-,182

-,264

-,077

n302

,506

,089

-,263

-,122

n847

,500

-,179

,496

-,135

n182

,491

-,245

,017

,197

n355

,470

-,083

-,312

-,088

n187

,450

-,163

-,142

-,362

n266

,422

-,184

-,176

-,099

n640

,520

,637

,032

-,108

n628

,480

,600

,035

-,011

n655

,465

,594

-,009

-,041

n385

,427

,539

-,093

,065

n890

,242

,456

-,112

,244

n548

,390

,041

,447

-,369

Variable Principal Normalization.

Rotated Component Loadingsa

Dimension

1

2

3

4

n195

,717

-,036

,070

,053

n248

,650

,037

,166

,024

n260

,609

,137

,087

,036

n325

,589

,098

,044

,122

n187

,575

,008

-,066

,213

n238

,574

,033

,188

,027

n355

,551

,109

,130

-,036

n302

,514

,278

,080

,010

n266

,477

-,006

,141

,076

n640

,207

,782

-,006

,186

n628

,149

,738

,066

,141

n655

,177

,726

,029

,111

n385

,169

,668

,104

-,007

n890

,005

,530

,185

-,151

n629

,223

,523

,199

,271

n291

,214

,104

,658

,287

n289

,285

,098

,623

,088

n270

,273

,129

,591

,003

n189

,396

,069

,547

,129

n182

,321

-,018

,461

,157

n295

,384

,223

,459

-,062

n288

,280

,153

,453

,140

n847

,181

,019

,233

,678

n362

,202

,048

,406

,662

n818

,209

,096

,327

,658

n548

,162

,162

-,078

,657

n361

,414

,183

,117

,562

Variable Principal Normalization.

a. Rotation Method: Quartimax with Kaiser Normalization. Rotation failed to converge in 5 iterations. (Convergence = ,000).

Component Transformation Matrixa

Dimension

1

2

3

4

1

,690

,396

,470

,382

2

-,279

,916

-,259

-,127

3

-,540

-,013

,128

,832

4

-,393

,063

,834

-,382

Variable Principal Normalization.

a. Rotation Method: Quartimax with Kaiser Normalization.

Round 2

Reviewer 1 Report

The reviewer would like to thank the authors to consider the entire comments. The paper is suitable for publication.